# Optimization of Cold Metal Transfer-Based Wire Arc Additive Manufacturing Processes Using Gaussian Process Regression

**Seung Hwan Lee**

School of Aerospace and Mechanical Engineering, Korea Aerospace University, 76 Hanggongdaehang-ro, Deokyang-gu, Goyang-si, Gyeonggi-do 10540, Korea; seunglee@kau.ac.kr; Tel.: +82-2-300-0106

**Abstract:** Wire and arc additive manufacturing (WAAM) is among the most promising additive manufacturing techniques for metals because it yields high productivity at low raw material costs. However, additional post-processing is required to remove redundant surface material from components manufactured by the WAAM process, and thus the productivity decreases. To increase productivity, multi-variable process parameters need to be optimized, including thermo-mechanical effects caused by high deposition rates. When the process is modeled, deposit shape and productivity are challenging to quantify due to uncertainty in multiple variables of the complicated WAAM process. Therefore, we modeled the WAAM process parameters, including uncertainties, using a Gaussian process regression (GPR) method, thus allowing us to develop a WAAM optimization model to improve both productivity and the quality of the deposit shape. The accuracy of the optimized output was verified through a close agreement with experimental values. The optimized deposited material had a wide effective area ratio, small height differences, and near 90° deposition angle, highlighting the usefulness of the GPR model approach to deposit nearly ideal material shapes.

**Keywords:** wire arc additive manufacturing (WAAM); buy-to-fly ratio (BTF); cold metal transfer (CMT); Gaussian process regression (GPR); parameter optimization

## 1. Introduction

Additive manufacturing (AM) for metals has been a topic of interest in recent years as a means to improve the productivity of manufacturing processes in various industries, including in the aerospace, shipbuilding, and automotive industries. AM has higher productivity and manufacturing flexibility than that in existing casting or forging processes [1–5].

Among various AM processes, wire arc additive manufacturing (WAAM) involves the deposition of a metal wire via an electric arc heat source. A typical welding machine can be used as a heat source in the WAAM process, which lowers the initial investment costs compared to powder-type AM processes, in which a laser or electron beam is used. Moreover, the cost of wire-type raw materials is approximately 10% of the cost of powder-type raw materials [6]. Therefore, the WAAM process yields a high deposition rate at a low cost, thus making it highly preferred in processes for manufacturing large components using expensive materials.

The components deposited using the WAAM process cannot be used in the deposited state due to redundant deposited areas. Moreover, the process requires post-processing such as milling or grinding, according to the specifications of the final component. Therefore, if post-processing can be minimized, the productivity of the WAAM process will be increased. The productivity of the WAAM process is evaluated using the buy-to-fly (BTF) ratio to include post-processing [7,8]. The BTF ratio is the

proportion of the quantity of raw materials removed for the final shape of the product and the quantity of raw materials supplied.

Several researchers have conducted experimental studies to reduce the quantity of raw materials removed to shape the final product by reducing surface waviness and roughness. Xiong et al. [9] confirmed that the surface waviness of a wall component decreased when the average arc current increased during a gas metal arc welding (GMAW) WAAM process using low-carbon steel wire. The waviness refers to repeated unevenness on the surface that resembles waves. Therefore, the reduction of waviness indicates improved productivity. Wang et al. [10] deposited a Ti-6Al-4V straight wall with a length of 975 mm using a gas tungsten arc welding (GTAW) WAAM process and repeatedly controlled the thermal boundary condition of the molten pool, thereby creating periodic and even surface waviness to increase productivity. Dinovitzer et al. [11] examined the effects of process parameters on the quality of the deposited shape using a specimen manufactured by depositing Hastelloy X alloy wire on 304 stainless steel in a TIG-based WAAM process. They reported that the travel speed and current had the most significant impact on the quality of surface roughness.

However, the surface waviness and roughness in these reports mostly result from cross-sectional images of the deposit. However, the cross-section is not enough to determine the productivity of the overall volume of the deposit. Therefore, additional parameters are needed to define overall uniformity, which is directly related to productivity. Specifically, horizontal uniformity can be expressed by the height difference between both ends of a layer, while vertical uniformity can be expressed by the deposit angle, based on the differences between the widths of the adjacent top and bottom layers. The following studies have been conducted to better understand the effects of WAAM process parameters on the horizontal uniformity of the deposit. Ou et al. [12] developed a three-dimensional heat transfer and fluid flow model of WAAM for H13 tool steel deposits and conducted a study on the uniformity of the resulting deposits. They calculated the deposit shape and size depending on arc power, travel speed, and the wire feed rate. They reported that the horizontal uniformity could be improved by controlling travel speed and the wire feed rate. Yehorov et al. [13] studied the effects of travel speed, the wire feed rate, and current on multi-layer single pass walls made from an Al–Mg alloy using a GMAW-based WAAM process. They report that higher horizontal uniformity at both wall ends was maintained by air cooling to control uneven heat concentration.

As seen from the above literature, although the productivity of the process should be assessed for the entire deposit volume, most researchers have focused on cross-section surface waviness or fractional experiments on vertical or horizontal depositions. The main reason for this is that, in addition to various process variables, the heat transfer affects the deposit shape. Moreover, maintaining volumetric uniformity is challenging while also leaving the shape of the cross-section unchanged. In conclusion, analyzing the correlation between the parameters of the WAAM process and the volume uniformity of the deposit is necessary to find the optimal combination of process parameters.

However, parameter optimization for the WAAM process is difficult because the WAAM is a multi-variable process. Therefore, modeling of the process cannot be derived analytically because the uncertainty of various process parameters can cause distorted variations in the deposition outcome. Accordingly, to optimize the deposition process, induced distortions, related noisy observations, and uncertainties must be considered. Gaussian process regression (GPR) is the best candidate because GPR is a non-parametric approach which can provide predicted values of unobserved points and more reliable prediction performance than other metamodels with a relatively small sample size [14]. Due to these advantages, GPR is applied widely in various engineering problems, including modeling geographical terrains [15], dynamical systems [16], non-linear systems [17], complex environments [18], and sensor networks, as well as in predicting the performance of arc welding [19], friction stir welding [20], gas tungsten arc welding [21], and robotic welding [22], and predicting the porosity in metal-based AM [23].

In this study, we model the WAAM process using the GPR method to optimize metal deposition in terms of the entire volume. The output data are evaluated based on the change in three major process

parameters: the wire feed rate, travel speed, and interpass time. The effective area (EA) ratio, which represents the quality of the cross-section, and the height difference and deposit angle, which represent the uniformity of the deposit, are used to analyze the deposit quality. A metamodel based on GPR is formed based on output data acquired from these results. The output data from the variation in major process parameters are included in the cost function for the optimization.

## 2. Wire Arc Additive Manufacturing (WAAM)

### 2.1. Setup of the WAAM Process

Figure 1 shows a schematic diagram of the experimental equipment used in the WAAM process. A TPS 4000 welder (FRONIUS, Wels, Austria) was used as a deposition heat source via a cold metal transfer (CMT) method that enables deposition with low heat input compared to other arc heat sources. A 200 mm × 80 mm × 15 mm sample of AH36 (Dongkuk Steel, Seoul, Republic of Korea) was cut and prepared as a substrate. Both ends of the substrate were clamped with the XY-axis stage to prevent distortion. The 10 wall layers were deposited on the substrate using a 1.2 mm M-316L (KISWEL, Seoul, Republic of Korea) wire made of STS 316L. Table 1 lists the chemical composition of the substrate and wire.

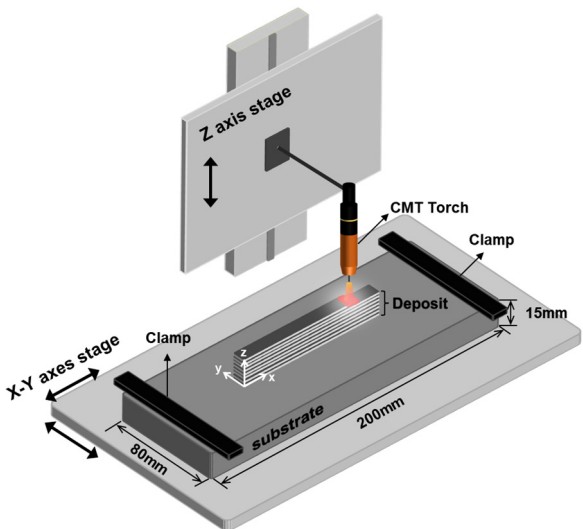

**Figure 1.** A schematic diagram of cold metal transfer (CMT) wire arc additive manufacturing (WAAM) equipment.

To control the deposition path and process automation in the WAAM process, we used an XYZ triaxial automated motion stage (Six Degrees Inc., Uiwang, Republic of Korea). The triaxial automated motion stage consists of an XY-axis stage on which the deposition specimen is secured and a Z-axis stage with a CMT torch. The WAAM process experiments are conducted in a zigzag direction until 10 layers of material 100 mm in length in the X-axis direction have been deposited. As the deposition begins, the XY-axis stage moves 100 mm at the experimental travel speed in the X-axis direction, after which air cooling is maintained during the interpass period. Subsequently, after the set interpass time has elapsed, the stage moves vertically to deposit the next layer at the proper height, and the torch moves to begin the next layer. After that, the stage then moves 100 mm in the opposite direction for the deposition of the next layer, after which the air cooling is again maintained during the interpass time. This process is repeated until 10 layers are deposited.

**Table 1.** Chemical composition of the wire (STS 316L) and substrate (AH36).

| Materials | Element (wt. %) | | | | | | | | | |
|---|---|---|---|---|---|---|---|---|---|---|
| | C | Si | Mn | P | S | Cu | Ni | Cr | Mo | V |
| Wire (STS316L) | 0.01 | 0.59 | 1.53 | 0.027 | 0.001 | 0.17 | 11.55 | 18.56 | 2.53 | |
| Substrate (AH36) | 0.15 | 0.24 | 1.19 | 0.012 | 0.003 | 0.022 | 0.011 | 0.02 | 0.001 | 0.001 |

The process parameters of the WAAM deposition are shown in Table 2. The 10 layer deposition experiment was repeated three times at four different levels, for each combination of process parameters. The average thickness of a single layer varies from 1.2 to 2.15 mm depending on the process parameters.

**Table 2.** Process parameters of the WAAM deposition.

| Number of Layers | Wire Feed Rate (m/min) | Travel Speed (m/min) | Interpass Time (s) |
|---|---|---|---|
| 10 | 2.6, 3.5, 4.4, 5.2 | 0.25, 0.50, 0.75, 1.00 | 0, 10, 30, 60 |

In the CMT welder, which uses a unity control system, current and voltage are synchronized and controlled simultaneously according to the wire feed rate. The wire feed rate is proportional to current and voltage; thus, an increase in the wire feed rate indicates an increase in heat input. In this study, the wire feed rate changes from 2.6 to 5.2 m/min, the corresponding voltage and current change from 10.7 V and 100 A to 12.5 V and 160 A.

The travel speed is an essential factor that determines the heat input and production speed of the deposits. Heat accumulation occurs due to the heat input generated when a layer is deposited in the multi-layer WAAM process. Thus, the interpass time has a significant impact on the occurrence of heat accumulation. The amount of heat accumulation varies by interpass time, which affects the solidification parameters (such as cooling rate), thereby ultimately influencing the deposited shape.

The micro Vickers hardness was measured to examine the mechanical properties of each layer of deposits. The specimen was cut from the center in the X-axis direction, and a cross-sectional specimen in the YZ-plane was created to measure the hardness of each layer. A load of 1000 gf was applied to the cross-sectional specimen for 10 s of dwell time using a hardness tester, HM-102 (Mitutoyo, Kawasaki, Japan). The hardness was measured three times at 0.5 mm intervals in the horizontal direction (Y-axis) from the center of each layer.

Macro images of the YZ cross-sections were captured using a BX51M (Olympus, Tokyo, Japan) light microscope (LM). The obtained LM images were then converted to binary images using MATLAB, and the EA ratio, height difference, and deposit angle were measured using the binary images.

## 2.2. Characteristic Parameters for the Deposit Shape

The micro Vickers hardness of the deposited specimens according to four different levels of interpass time is shown in Figure 2. The hardness is the mean of the three measurements. The cooling effect through the substrate was fast across all interpass times in the first and second layers, which are located close to the substrate when compared to other layers. Consequently, the refined structure caused by the increased cooling rate leads to higher hardness, as shown in Figure 2. Previous studies reported that the hardness of STS 316L components deposited using a WAAM process was between 170 and 190, except bottom layers, where the mechanical properties are not constant [24,25]. Therefore, substrates are removed, and only the deposits from above a certain height are used at actual worksites [26]. Thus, we only examined the deposited shape above the fourth layer of the deposits.

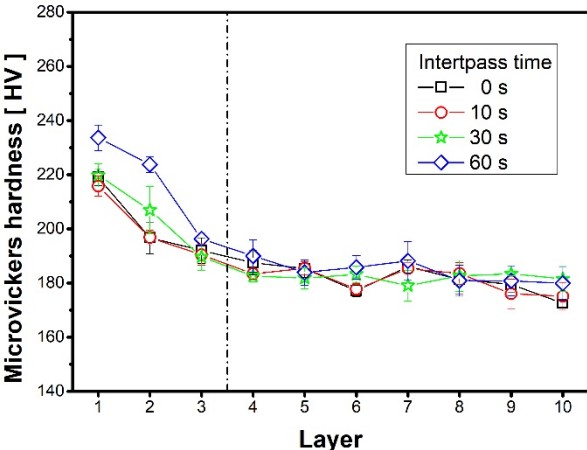

**Figure 2.** The micro Vickers hardness of deposits in each layer with varying interpass time.

As mentioned in Section 2.1, the LM images showing the cut deposit ware converted to a binary image, as shown in Figure 3a. The pixels of the binary image are expressed in 0 (black) and 1 (white). Using the number of pixels, the total area and the EA were calculated, as shown in Figure 3b. Here, the total area refers to the entire area of the deposits, excluding the substrate, and the EA is the largest rectangle with the gray color in the area above the fourth layer. The EA ratio is defined as the difference between the total area and the EA, divided by the total area. If the EA ratio is uniform throughout the entire deposit, which corresponds to the cross-section, it is identical to the BTF ratio, which represents the entire deposit. Therefore, in a general case of non-uniform deposits, we need to define the deposit angle and height difference for optimization. After creating a straight line connecting point A, which represents the valley of waviness in the third and fourth layers, and point B, which represents the valley of waviness in the ninth and tenth layers, the deposit angle was defined as the angle between the straight and horizontal lines, as shown in Figure 3b. The height difference was measured at both ends of the deposit, as shown in Figure 3c.

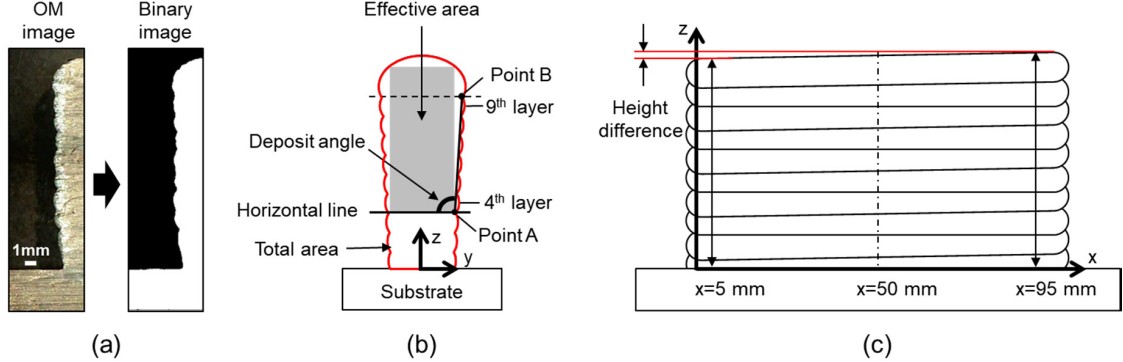

**Figure 3.** (**a**) Light microscope (LM) and converted binary images, (**b**) schematic showing EA and deposit angle, and (**c**) schematic showing height difference measured at both ends.

### 2.3. Effect Of Process Parameters on Deposit Shape

In this section, we examine the effects of the wire feed rate, travel speed, and interpass time on the deposited shape. The deposit shape is set as an objective function for optimization modeling, which will be discussed in Section 3. Among the three process parameters, one independent parameter is changed in the experiment to examine its effect on the objective function.

### 2.3.1. Comparison of Deposit Shape Depending on the Wire Feed Rate

The correlation between the wire feed rate and the EA ratio, height difference, and deposit angle is shown in Figure 4. The experiment was conducted by setting the travel speed and interpass time to 0.5 m/min and 30 s, respectively, and increasing the wire feed rate from 2.6 to 5.2 m/min at increments of 0.9 m/min for four levels. As shown in Figure 4a, the EA ratio increases until the wire feed rate reaches 4.5 m/min before decreasing. The height difference significantly increases after the wire feed rate exceeds 3.5 m/min, as shown in Figure 4b, while the deposit angle increases proportionally with the wire feed rate (Figure 4c). The heat input increases with the wire feed, which in turn deepens the surface waviness. Furthermore, as the heat input increases, the deposits become irregular at both ends due to spattering. Consequently, the height difference between the two ends significantly increases as the wire feed rate also increases. As the height difference becomes smaller, the changes in contact tip-to-work distance also becomes small while subsequent layers are deposited, indicating a stable process.

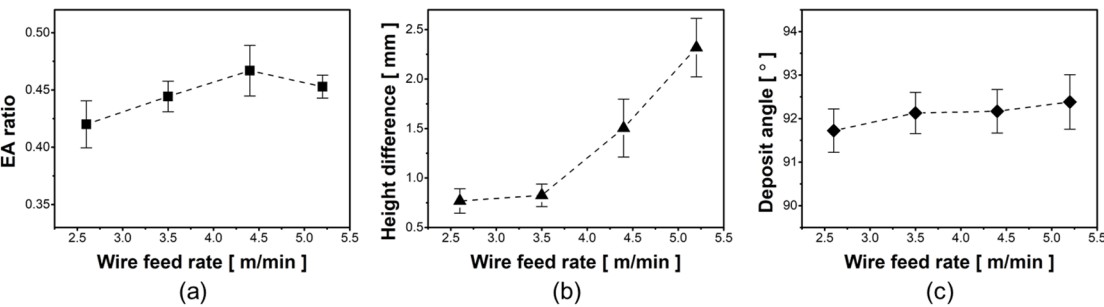

**Figure 4.** Experimental results depending on the wire feed rate: (**a**) the effective area (EA) ratio, (**b**) height difference, and (**c**) deposit angle.

### 2.3.2. Comparison of Deposit Shape Depending on Travel Speed

The variation in deposit shape when the travel speed was increased from 0.25 to 1.0 m/min at an increment of 0.25 m/min is shown in Figure 5. In this experiment, the wire feed rate and interpass time were set to 3.5 m/min and 30 s, respectively. In Figure 5a, the correlation between the travel speed and the EA ratio is shown. An increase in the travel speed led to the irregular bead shape induced by the unstable arc. Accordingly, the EA ratio increases as the effective area decreases. However, when the travel speed is 1.0 m/min, a humping layer was generated, and the humping reduces the total area as well as the EA ratio. The height at both ends, shown in Figure 5b, is the greatest when the travel speed is 0.25 m/min, at which point the large heat input generates spattering, causing non-uniform deposition shape. The height difference is the smallest at 0.5 m/min and gradually increases with the travel speed. Finally, the deposit angle is inversely proportional to the travel speed because the speed of molten metal spread slows down when the heat input decreases, as shown in Figure 5c.

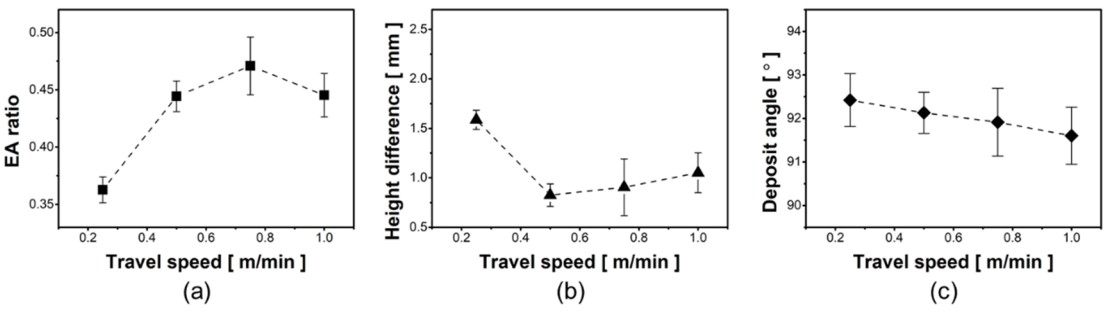

**Figure 5.** Experimental results depending on travel speed: (**a**) the EA ratio, (**b**) height difference, and (**c**) deposit angle.

2.3.3. Comparison of Deposit Shape Depending on Interpass Time

The correlation between the interpass time and the EA ratio, height difference, and deposit angle are shown in Figure 6. The wire feed rate and travel speed are set to 3.5 m/min and 0.5 m/min, respectively, while the interpass time changes from 0 s to 60 s. The fixed wire feed rate and travel speed signify uniform heat input, allowing the effect of heat accumulation due to interpass time on the deposited shape to be examined. As shown in Figure 6a, the EA ratio is the smallest when the interpass time is 0 s, where deposition occurs continuously without idle time between layers, and the molten pool flows down due to heat accumulation effect caused by slow cooling rate. Accordingly, the EA ratio is small because the molten pool fills the valley of the surface waviness, thus reducing the surface waviness. The EA ratio increases with interpass time and then decreases again when the interpass time exceeds 30 s, because the decrease in the deposit angle has more effect than the increase in the surface waviness depending on interpass time. The height difference shown in Figure 6b decreases until the interpass time reaches 30 s and remains consistent afterward. Changes in the deposit angle are shown in Figure 6c, which decreases as the interpass time increases. Bottom layers of the deposits tend to have a narrow and tall shape because the cooling rate is fast due to extensive heat dissipation through a substrate. Conversely, heat accumulates in the top layers due to repeated deposition, and the cooling rate reduces as the heat accumulation increases. Therefore, top layers tend to have a wide and low shape because the amount of lateral flow is high until the layers solidify. This phenomenon becomes more evident with high heat accumulation, which indicates that the deposit angle is large.

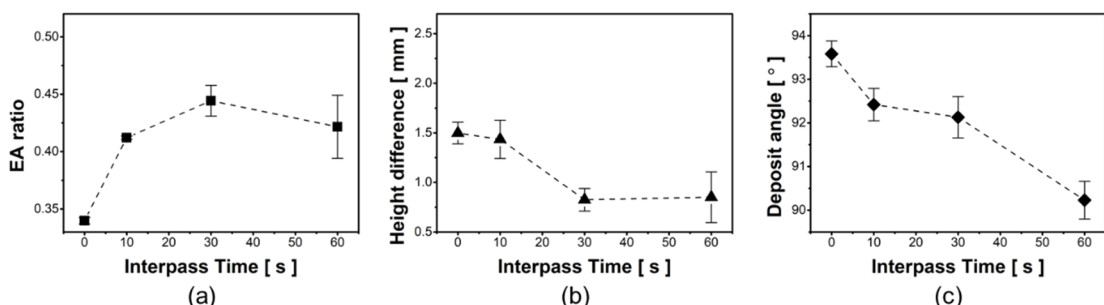

**Figure 6.** Experimental results as a function of interpass time: (**a**) the EA ratio, (**b**) height difference and (**c**) deposit angle.

## 3. Optimization Methodology

Modeling of the WAAM process cannot be derived analytically because the uncertainty of various process parameters can cause distorted variations in the deposition outcome. Therefore, a non-parametric approach which a is data-driven approach is suitable for this process. In this section, we explain how GPR is applied to predicting WAAM process parameters. After introducing GPR, the experimental data are used in developing a model for defining the EA ratio, height difference, and deposit angle. Subsequently, the model is applied to the WAAM parameter optimization using a cost function.

### 3.1. Gaussian Process Regression Modeling

A Gaussian process is a stochastic process expressed in terms of mean and covariance [27]. As explained above, wire feeding speed, travel speed, and interpass time were used as input parameters for the GPR model. Depending on the change in these input parameters, the EA ratio, height difference, and deposit angle were affected. The cost function was then calculated using the combination of these output values. The input parameters that minimize the developed cost function will be the optimal values.

For GPR, the following data set is generated using the WAAM experiments.

$$S_k = \left\{ \left( \begin{array}{cc} \mathbf{x}_i & y_i^j \end{array} \right); i = 1, \cdots, n \right\}, j = 1, \cdots, m \tag{1}$$

Here, $\mathbf{x}_i \in \mathbb{R}^d$ is the experiment setting input and $y_i^j \in \mathbb{R}$ is the experiment output for which the distribution of each variable is assumed to be unknown. We prepared 12 data sets ($n = 12$) for the experiment, and each data set has three parameters ($d = 3$) and three different measurements ($j = 3$).

The GPR model can be expressed in a mathematical model as follows:

$$y = h(\mathbf{x})^T \boldsymbol{\beta} + f(\mathbf{x}) \tag{2}$$

Here, $f(\mathbf{x})$ has a mean value of 0 from the Gaussian process with the covariance function $k(\mathbf{x}, \mathbf{x}')$. In this study, $h(\mathbf{x})$ is a set of basis functions of an input vector $\mathbf{x}$ in the $\mathbb{R}^d$ dimension, while $\boldsymbol{\beta}$ is a coefficient vector of the basis function [27,28].

The covariance function $k(\mathbf{x}, \mathbf{x}')$ is affected by $\theta$, and is expressed as $k(\mathbf{x}, \mathbf{x}'|\theta)$. The prediction of new results from GPR is determined by the coefficient $\boldsymbol{\beta}$, kernel coefficient $\theta$ of the covariance function $k(\mathbf{x}, \mathbf{x}'|\theta)$, and the noise distribution $\sigma^2$ [27]. For the covariance function, we used the automatic relevance determination squared exponential kernel, as follows:

$$k(\mathbf{x}_i, \mathbf{x}_j) = k(\mathbf{x}_i, \mathbf{x}_j|\theta) = \sigma_f^2 exp\left[ -\frac{1}{2} \sum_{k=1}^{d} \frac{\left( \mathbf{x}_{i,k} - \mathbf{x}_{j,k} \right)^2}{\sigma_m^2} \right] \tag{3}$$

Here, $\sigma_f$ is the standard deviation of the signal, while $\sigma_m$ is a coefficient that standardizes how far off the predicted input values are from the sample values $(1, 2, \cdots, d)$ [28].

The coefficient $\theta$ is determined as follows:

$$\theta_m = log\sigma_m, \; for \; m = 1, 2, \cdots, d \theta_{n+1} = log\sigma_f$$

GPR can select several basis functions such as linear and quadratic functions. However, in our problem, there is no analytic information of the WAAM process. Thus, it is difficult to apply the parametric approach. In this study, therefore, the basis function $h(\mathbf{x})$ is an n-by-1 vector of ones, while the arbitrary value $\mathbf{x}_{new}$ and the predicted value $\hat{y}(\mathbf{x}_{new})$ are defined as follows.

$$\hat{y} = h(\mathbf{x}_{new})^T \hat{\beta} + K^T(\mathbf{x}_{new}, \mathbf{X}) \left( K(\mathbf{X}, \mathbf{X}) + \sigma^2 I_n \right)^{-1} \left( Y - H\hat{\beta} \right) \tag{4}$$

In this equation, $H$, $K(\mathbf{x}_{new}, \mathbf{X})$, and $K(\mathbf{X}, \mathbf{X})$ are defined as follows:

$$H = \begin{bmatrix} h\left(\mathbf{x}_1^T\right) \\ \vdots \\ h\left(\mathbf{x}_n^T\right) \end{bmatrix}, \quad K(\mathbf{x}_{new}, \mathbf{X}) = \begin{bmatrix} k(\mathbf{x}_{new}, \mathbf{x}_1) \\ \vdots \\ k(\mathbf{x}_{new}, \mathbf{x}_n) \end{bmatrix}, K(\mathbf{X}, \mathbf{X}) = \begin{bmatrix} k(\mathbf{x}_1, \mathbf{x}_1) & \cdots & k(\mathbf{x}_1, \mathbf{x}_n) \\ \vdots & \ddots & \vdots \\ k(\mathbf{x}_n, \mathbf{x}_1) & \cdots & k(\mathbf{x}_n, \mathbf{x}_n) \end{bmatrix} \tag{5}$$

where the predicted value $\hat{\beta}$ is determined according to the following equation:

$$\hat{\beta} = \left\{ H^T \left( K(\mathbf{X}, \mathbf{X}) + \sigma^2 I_n \right)^{-1} H \right\}^{-1} H^T \left( K(\mathbf{X}, \mathbf{X}) + \sigma^2 I_n \right)^{-1} Y \tag{6}$$

The value $\sigma$ is determined by the experimental results, and the vector $Y$ uses each measurement value $y$ as its scalar component [29].

### 3.2. GPR models for WAAM Process Parameters

Three output values for optimization were predicted using the GPR model. The experimental data from Section 2.3 are given as training data sets, and the predicted results are calculated, as shown in Figure 7. The red asterisks mark the average of three experiments conducted for 12 data sets, while the lines are the regression curves predicted by the GPR model. As shown in Figure 7a, the EA ratio of the GPR model increases until the third level and then decreases in the fourth level. The EA ratio is smallest when the travel speed is 0.25 m/min. As shown in Figure 7b, the height difference of the GPR model decreases when the wire feed rate is 2.5 m/min, reaching a minimum value at 3.0 m/min, before subsequently increasing to reach a maximum value at 5.2 m/min. The height difference is the smallest when the travel speed is approximately 0.6 m/min and then increases with the travel speed. Furthermore, the height difference decreases as the interpass time increases and reaches a minimum value when the interpass time is approximately 45 s. Afterward, the height difference increases again until the interpass time reaches 60 s. This trend between 30 and 60 s is different from the results described in Section 2, because of the standard deviation of the experimental data, which is expressed as $\sigma_f$ in Equation (3). This deviation strongly affects the multi-variate GPR. As shown in Figure 7c, the deposit angle increases proportionally with the heat input, as determined by the wire feed rate or travel speed. However, it exhibits relatively more significant changes than other process parameters when the interpass time increases. The deposit angle decreases from 0 to 20 s, increases until 30 s, and then decreases again until 60 s. In the GPR model, the strong dependence of the deposit angle on the interpass time indicates an additional influence on heat accumulation.

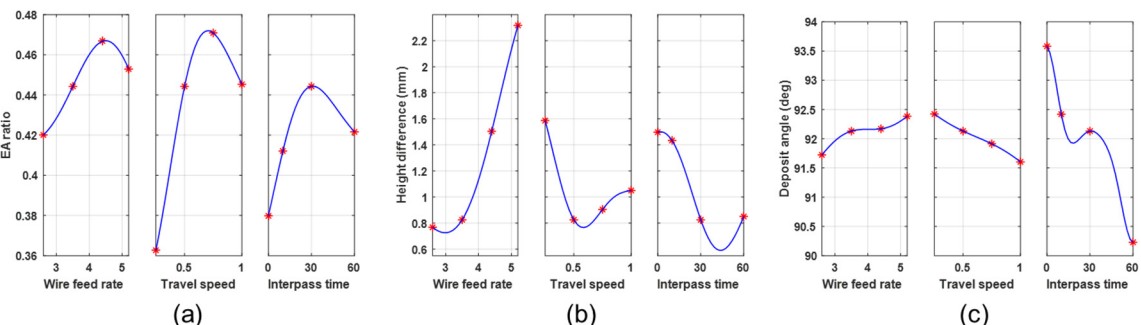

**Figure 7.** Output values of the Gaussian process regression (GPR) model (red asterisk indicates the experimental data).

To reduce material waste, the EA ratio, height difference, and deposit angle must be optimized. EA ratio minimization can be accomplished by reducing the surface waviness of the deposits, while the height difference and deposit angle must be reduced to improve uniformity in both horizontal and vertical directions. Using these results, the optimization of the WAAM process parameters using the predicted GPR model will be discussed in the following section.

### 3.3. Optimization of the WAAM Parameters

The input variables for finding the optimal parameters for the WAAM process are defined as follows:

$$\mathbf{x} = \left[ \mathbf{v}_f, \; \mathbf{v}_t, \; \mathbf{t}_{ip} \right] \tag{7}$$

Here, $\mathbf{v}_f \in [\, 2.6 \; 3.5]\; m/min$, $\mathbf{v}_f \in [0.25 \; 1.0]\; m/min$, and $\mathbf{v}_f \in [\, 0 \; 60]\; s$, while $\mathbf{v}_f$ and $\mathbf{v}_t$ represent the wire feed rate and travel speed. $\mathbf{t}_{ip}$ represents the interpass time. The range of input variables is used as a constraint in the optimization process. The intermediate result vector $g(\mathbf{x})$ which is the function of $\mathbf{x}$ is defined as follows:

$$g(\mathbf{x}) = [EA, \; D_h, \; \theta_d \,] \tag{8}$$

Here, *EA* is the EA ratio, $D_h$ is the height difference, and $\theta_d$ is the deposit angle. The cost function can be defined using the output data, and the optimization of parameters is set to minimize the cost function as follows:

$$\min\{J(\mathbf{x})\} \tag{9}$$

Here,

$$J(\mathbf{x}) = \frac{1}{3}[1\ 1\ 1] \times g(\mathbf{x})^T \tag{10}$$

The same weighted value is applied to all three input variables because the three outputs consequently represent three-dimensional geometrical effects. This weighted value can be adjusted as a user gives priority to specific shapes or characteristics. The weight value can be combined in various ways to obtain different characteristics accordingly. However, studies on weight values should be examined further in future research as the scope of this inquiry is broad.

Sequential quadratic programming was used as an algorithm to find the optimal values, with parameters calculated as shown in Figure 8a below.

$$\mathbf{x}^* = [3.1002\ 0.3522\ 57.972]^T$$

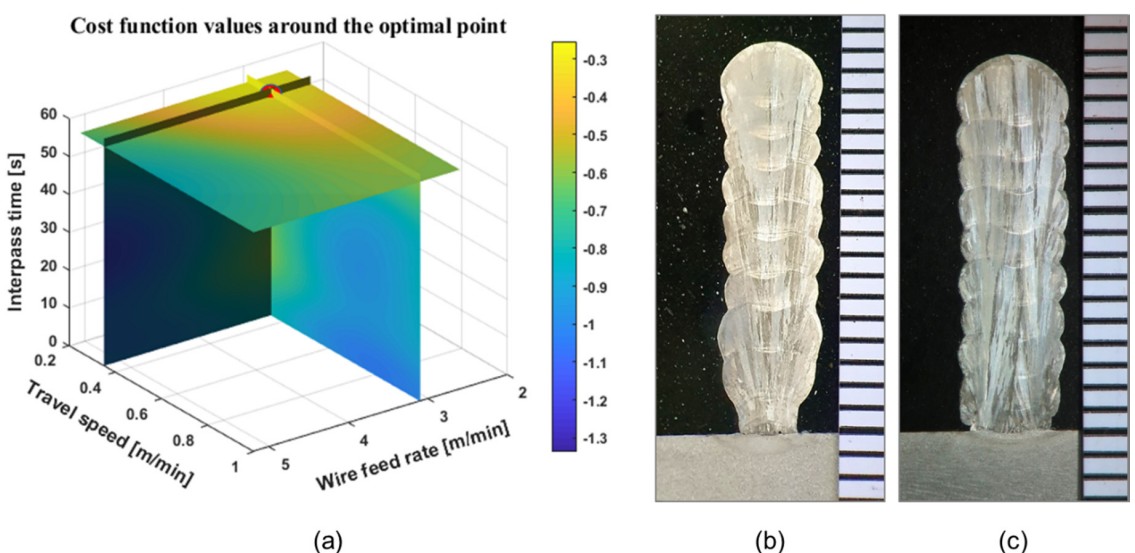

**Figure 8.** (**a**) The cost function and (**b**) fusion zone shapes before optimization and (**c**) after optimization.

To verify the possibility of hitting a local minimum, 20 different initial points were arbitrarily selected and executed for comparison. All the comparison results represent the final output values, limited by the tolerance threshold of $1 \times 10^{-8}$, which are delivered to the optimizer as parameters.

### 3.4. Verification of Parameter Optimization

The effects of three parameters on the cost function are shown in Figure 8a. The x, y, and z axes represent the WAAM process parameters, while the cost function values are represented by the color bar. If the color gradient converges to a red point in Figure 8a, this indicates no local optimal points. The experimental validation was performed using the optimization parameters obtained from the GPR model, and the cross-sectional shape before and after the optimization is expressed, as shown in Figure 8b,c. Regarding the experimental conditions in Figure 8b, the wire feed rate was 3.5 m/min, travel speed was 0.5 m/min and interpass time was 30 s.

Table 3 shows the experimental mean values with standard deviation and the prediction from the GPR model with the optimal parameters. The EA ratios are 0.3905 and 0.4069, the height differences are 1.0197 and 1.0640 mm, and the deposit angles are 90.3948° and 90.3023°, respectively, with

corresponding errors of 4.2%, 4.3%, and 0.1%. As the number of experimental data increases, the experimental mean value will be closer to the predicted value of the GPR model. The deposit shape of the WAAM experiments conducted using the optimized GPR model had a wide EA, a small height difference between both ends, and a deposit angle close to 90°. Consequently, after optimization, the BTF ratio was improved from approximately 0.49 to 0.43. These results highlight the effectiveness of using a GPR model to optimize WAAM process parameters.

**Table 3.** Experimental values and predictions from the GPR model with the optimal parameters.

|  | EA Ratio | Height Difference [mm] | Deposit Angle [°] |
|---|---|---|---|
| GPR | 0.3905 | 1.0197 | 90.3948 |
| Experimental mean value (STD) | 0.4069 (0.008) | 1.0640 (0.09) | 90.3023 (0.57) |

## 4. Conclusions

In this study, WAAM process deposition parameters were optimized to improve the quality of the deposit shape and the productivity of the WAAM. The wire feed rate, travel speed, and interpass time were modeled using a GPR method. The productivity of the WAAM process is defined by output data, including the EA ratio, height difference, and deposit angle, which represent the deposit shape in three dimensions. When the modeling of GPR was performed, the output data variation effects were taken into consideration and used to calculate the cost function. The output optimized through GPR modeling was verified for prediction accuracy by comparing with experimental results gathered under the same process conditions. Similarities between the GPR model and experimental values suggest that GPR can accurately model the WAAM process, despite large uncertainties and noise. These uncertainties occur due to the complicated effects that various process parameters have on deposition in the WAAM process. Even when a small number of experiments are conducted, optimization based on models like GPR is more useful for finding optimal parameters than trusting an operator's experience or intuition. This approach can be used to improve the performance of WAAM-based manufacturing processes involving diverse materials and shapes.

**Funding:** This work was supported by Korea Aerospace University (No. 2019-01-006) and a National Research Foundation of Korea (NRF) grant funded by the Korea government (MSIT) (No. 20201C1C1009519).

**Conflicts of Interest:** The authors declare no conflict of interest.

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
