# Peer review of "Optimization of Cold Metal Transfer-Based Wire Arc Additive Manufacturing Processes Using Gaussian Process Regression"

_metals, doi:10.3390/met10040461_

Round 1
Reviewer 1 Report
This article deals with multivariant wire and arc additive manufacturing process and reports the optimization results of WAAM technological parameters calculated by Gaussian method. Authors presented well-documented shape analysis of walled samples vs. WAAM parameters, i.e. wire feed rate, travel speed, interpass time. Based on obtained results the Gaussian regression process was conducted and finally proved by real experiment.
Summarizing, the article is written in logical way and good in English. Moreover the presented information can be very useful for science and industry additive manufacturing communities. I recommend this paper publishing in Metals journal after some major/minor corrections listed below:
- The 2.1 section concerns strictly WAMM process not microstructure!! Moreover the samples microstructure was not investigated!! You should also remember that “microstructure” term should be always used in singular form.
- Please add information how was prepared substrate for WAAM process.
- The WAAM process should be precisely parametrized – What were voltage and current of WAAM process?
- Did you use STL model? If yes please add suitable information.
- What was the thickness of single layer?
- “This process is repeated until the desired thickness is reached.” – should be “This process is repeated until the desired sample height is reached. “desired” – it means how much?
- Table 1 – Did you measure chemical composition of wire and substrate? If not the reference/es should be given.
- “A pressure of 1000 gf was applied to…” should be “A load of 1000 gf was applied to…”
- Please use Light Microscopy term instead of Optical Microscopy – please check throughout the article.
- Figure 2 – Is it presented mean values of microhardness? Please explain it.
- Figure 4,5,6 – please change solid lines into dashed lines.
Author Response
I appreciate your valuable comments.
Please see the attachment.

Reviewer 2 Report
This paper presents experimental research in conjunction with empirical computational modelling research on the direct energy deposition technique for additive manufacturing of metals. This research is of significant importance to the body of knowledge in terms of a methodology for the optimisation of additive manufacturing process. It has clear potential to directly benefit the industry by helping the industry to improve the quality of it 3D builds. The literature of the paper is comprehensive and the presentation of the research methods and results is clear and sufficient. This paper can be accepted for publication is the authors can revise the manuscript according to the following specific comments of the reviewer.
- The title of section 2.1 has nothing to do with the microstructure of the weld. The title of this section should be something like “Setup of the WAAM process”
- The English of the manuscript needs to be improved. There are some inappropriate expressions like “An increased cooling speed decreases grain size” etc. The authors are advised to find a native speaker of English to help them do a proof reading for the manuscript of this paper.
- In relation to equation (2), what does it mean by “In this study, the basis function ℎ(?) is an n-by-1 vector consisting of 1s”? What is the notation s for? The n-by-1 vector consist of 1s what? The presentation of the research method of the Gaussian process regression needs improvement. For example, the authors should present briefly what this method is and overall how it works and then clearly introduce the related mathematical expressions for this method and explain all the related notations.
- In relation to Figure 7, what is the size of your training data set? Is it 12 like what you presented in the paragraph below Figure 7? This data set looks like too small for “training” a predictive model that was supposed to predict three results of the additive manufacturing simultaneously. What’s the authors comment on the size of their training data set?
Author Response

(The authors gave the same response as above.)

Reviewer 3 Report
Overall, paper is nicely written, the idea is novel, which provides potential solution for improving dimension accuracy for WAAM.
The justification for using GPR is not fully justified as the best method - only area needs to be improved.
Author Response
I appreciate your valuable comment.
Please see the attachment.

Round 2
Reviewer 1 Report
I have no more comments.